# Agro-Morphological and Cytogenetic Characterization of Colchicine-Induced Tetraploid Plants of *Polemonium caeruleum* L. (Polemoniaceae)

**DOI:** 10.3390/plants11192585

**Published:** 2022-09-30

**Authors:** Tatiana E. Samatadze, Olga Yu. Yurkevich, Firdaus M. Khazieva, Irina V. Basalaeva, Elena A. Konyaeva, Alla E. Burova, Svyatoslav A. Zoshchuk, Alexander I. Morozov, Alexandra V. Amosova, Olga V. Muravenko

**Affiliations:** 1Engelhardt Institute of Molecular Biology, Russian Academy of Sciences, 32 Vavilov St., Moscow 119991, Russia; 2All-Russian Institute of Medicinal and Aromatic Plants, 7 Green St., Moscow 117216, Russia

**Keywords:** *Polemonium caeruleum*, colchicine, diploid, autotetraploid, agro-morphological parameters, pollen, karyotype, chromosome, FISH, meiosis

## Abstract

*Polemonium caeruleum* L. (Polemoniaceae) is a valuable medicinal herb with a wide spectrum of biological activities. Under natural conditions, the productivity of this species is rather low. In this study, colchicine-induced tetraploid plants (2n = 4x = 36) of *P. caeruleum* were obtained, and for the first time, their morphological and cytogenetic characterization was performed. In the tetraploid plants, raw material productivity and also the content of triterpene saponins were significantly higher than in the control diploids. The analysis of chromosome behavior at meiosis and FISH chromosome mapping of 45S and 5S rDNA generally demonstrated stability of both genomes in the tetraploid plants. Based on chromosome morphology and distribution patterns of the studied molecular cytogenetic markers, all chromosome pairs in karyotypes were identified, and chromosome karyograms and idiograms of *P. caeruleum* were constructed. The revealed specific microdiagnostic characteristics of *P. caeruleum* (strongly sinuous cells and anomocytic stomata of the leaf epidermis, and also glandular hairs along the veins) could be useful for raw material identification. In the obtained tetraploids, the predominance of large stomata on the lower leaf epidermis was determined. The studied tetraploids can be used in various breeding programs to obtain high-quality pharmaceutical raw materials of *P. caeruleum*.

## 1. Introduction

Blue cyanosis (*Polemonium caeruleum* L., Polemoniaceae) is a perennial medicinal herb with a wide spectrum of biological activities, which allows this species to be used for the prevention and treatment of many diseases [1], and due to numerous application possibilities, this plant is important for traditional medicine [2]. *P. caeruleum* is widespread in Eurasia and North America, mostly growing along valleys and river banks, and also in meadows [3,4].

For medicinal purposes, the whole blue cyanosis plant is used. The roots of cyanosis contain saponins, anthocyanins and carotenoid pigments [5,6,7]. Flavonoids, carbohydrates, fatty acid esters, amino acids and carboxylic acids were found in the stems and leaves [5,8]. All these compounds have a wide range of activities, including antioxidant, antitumor, anti-inflammatory, antiatherogenic, diuretic, antifungal, antiviral, antiarrhythmic and detoxifying properties, resulting in a wide demand for blue cyanosis raw materials [5,6,7,9,10]. However, under natural conditions, the productivity of this species is reduced, and its harvesting is not effective enough [11].

Currently, artificial polyploidization is often used to develop high-yielding populations of plants rich in biologically active compounds [12]. Autopolyploidization of genomes of medicinal plants is an effective tool to increase the number and size of their vegetative and generative organs, plant biomass, content of secondary metabolites and biologically active compounds [13,14,15]. In particular, for colchicine-treated *Petunia hybrida* cv. Mitchell, increased concentration of major flavonols (quercetin-3-sophoroside) and also decreased content of minor flavonols (quercetin-3,7-diglucoside) have been reported [16]. Increased contents of bioactive ingredients have previously been revealed in a range of tetraploid plants, such as *Matricaria chamomilla* (apigenin) [17], *Artemisia annua* (artemisinin) [18], *Solanum tuberosum* (phenylpropanoids) [19] and *Hyoscyamus muticus* (scopolamine) [20].

In natural populations, blue cyanosis is a diploid species with 2n = 2x = 18 chromosomes [21,22,23]. At the same time, the development of autopolyploid plants of blue cyanosis could help solve the problem of raw material deficiency of this valuable medicinal plant. Colchicine is widely used to change the ploidy level in crop production [18,24,25,26,27]. This cytostatic agent is known to bind with tubulin dimers, which might prevent spindle formation, shorten the length of spindle fibers, and lead to temporary inactivation of the division spindle. Mitotic daughter chromosomes are thus prevented from separating, and all chromosomes are included in one nucleus with the double chromosome number [28].

The obtained polyploid plants usually have larger inflorescences, fruits, and pollen grains, and sometimes have shorter stems [12,29]. At the same time, mutagenic effects of colchicine can influence the plant genome structure causing physiological disturbances and also chromosome disorders [30,31]. For instance, various meiotic disorders, including uneven chromosome segregation and formation of uni-, tri- and tetravalents, might occur in the reduction division of colchicine-induced polyploids [27,32,33], which is a major hindrance to using such polyploid plants in plant breeding [12,34]. Therefore, a cytogenetic analysis of the colchicine-induced autopolyploids, as well as their monitoring in subsequent generations, is required to exclude plants with unbalanced or aneuploid genomes and chromosome disorders [28]. However, currently available karyological data on *P. caeruleum* are based on simple monochrome staining of chromosomes, and only the basic numbers of chromosomes and their sizes (ranging from 2 to 5 µm) are reported [21,22,23]. Modern molecular cytogenetic techniques will make it possible to obtain valuable information regarding the structural organization of the *P. caeruleum* genome, which is useful for breeding efforts focused on the development of new genetically stabilized populations or cultivars.

In the present study, we aimed to develop colchicine-induced tetraploid plants of *P. caeruleum* to improve the main characteristics of this valuable medicinal plant. Furthermore, we aimed to perform the comparative agro-morphological and cytogenetic characterization of the obtained tetraploids and the control diploid plants. Such an approach will make it possible to evaluate the stability of their genomes, which is essential for potential breeding programs.

## 2. Results

### 2.1. Agro-Morphological Characterization

To obtain polyploid plants, the apical meristem of shoots of two- and four-leaf plants was treated with different concentrations of aqueous colchicine (0.05%, 0.1%, 0.2%, and 0.5%). Our results demonstrated that colchicine can induce polyploidy in the studied *P. caeruleum* cultivar, but the efficiency of this induction was different. It was revealed that high concentrations of colchicine (0.5%) reduced the survival rate of the seedlings, and the induction of putative polyploids was also decreased. At colchicine concentrations of 0.05% and 0.1%, the seedlings almost did not differ biometrically from the control ones, so they were excluded from further study. The largest percent of putative polyploid plants (15%) was obtained after the treatment with 0.2% colchicine for 24 h (Figure 1).

At the stage of six–eight leaves, the colchicine-treated plantlets exhibiting morphological differences from the control diploids were tested for the number of chromosomes in karyotypes to confirm their tetraploid status. As a result, nine tetraploid plants were revealed. The following season, the screening of the obtaining tetraploid plants demonstrated their stable tetraploid nature. At the same time, they differed from the control diploids in terms of morphometric parameters, as well as with respect to productivity. In particular, the tetraploid plants were undersized when compared with the control (Figure 2, Table 1). Additionally, mixoploid root tissues were also detected; however, they were discarded from the experiment, as they had an unstable polyploid nature.

In tetraploid plants, leaf blades were dark green, and diversity in color and shape of inflorescences was also observed (Figure 3). The morphological differences between the control and colchicine-induced tetraploid plants were easily identified visually, and these characteristics were used for preliminary identification of tetraploids among the studied *P. caeruleum* plants (Table 1).

In tetraploid plants, the raw material productivity and seed yield exceeded (by 32% and 23%, respectively) those parameters in the control plants (Figure 4, Table 1). Weight of 1000 seeds (g) was 1.31 ± 0.116 (in diploids) and 1.54 ± 0.125 (in tetraploids). At the same time, tetraploid plants were characterized by lower sowing quality of seeds (germination energy and germination of seeds) compared to the control (Figure 4). Moreover, in tetraploid plants, the content of triterpene saponins in terms of β-escin determined in the rosettes of leaves and rhizomes with adventitious roots (23.58% and 20.36%, respectively) significantly exceeded those characteristics observed in the control plants (18.89% and 14.92%, respectively) (Figure 4, Table 1).

### 2.2. Cytogenetic Characterization

A comparative analysis of chromosome behavior during meiosis was carried out in the control diploids and colchicine-induced tetraploid plants. At metaphase I (M-I), regular meiotic chromosome behavior with normal chromosome disjunction was observed in most maternal pollen cells (MPC) of the studied plants. The diploid plants had nine bivalents and the polyploid plants had eighteen bivalents (Figure 5a,b). Additionally, several meiotic abnormalities including chromosomes lying outside the metaphase plate were revealed in the MPC of diploid plants (2.15%). In the cells of tetraploid plants, univalents and also various polyvalent chromosome associations (trivalents and quadrivalents) were detected (Figure 5c,d). The number of the revealed multivalents per cell was variable. In the colchicine-induced tetraploids, the quadrivalents (2.17 ± 1.13), trivalents (0.64 ± 0.22) bivalents (10.27 ± 2.04) and also univalent (1.62 ± 0.15) were revealed.

At anaphase I (A–I) stage, we observed a segregation of chromosomes 9:9 in the MPC of the control diploids. In the studied tetraploid plants, a segregation of chromosomes 18:18 was detected (Figure 5e,f). In addition, chromosome lagging and unequal distribution of genetic material were observed in the MPC of diploids (2.27%) and tetraploids (37.14%). At anaphase II (A-II), bridges, a perpendicular spindle orientation and also nonuniform chromosome distribution were detected in the MPC of diploid (1.68%), and tetraploid (21.76%) plants (Figure 5g,h). Apparently, such meiotic abnormalities eventually led to the presence of pentads in a small number of cells at the final stages of meiosis. However, mostly tetrads were observed at those stages (Figure 5i).

### 2.3. Karyotype Structure and FISH Analysis

The karyotype of the control *P. caeruleum* plants comprises 2n = 2x = 18 middle-sized chromosomes ranged from 3.6 to 5.0 µm (Figure 6a). Clusters of 45S rDNA were localized in the short arms of three satellite (SAT) chromosome pairs 3, 4 and 7. The secondary constrictions of chromosome pairs 3 and 4 were rather short unlike chromosome 7 having a long thread with a satellite. Bright 5S rDNA signals were detected on the short arms of chromosome pair 4, and also on the long arms of chromosome pair 8.

The karyotype of tetraploid *P. caeruleum* plants contained two similar genomes 2n = 4x = 36, and chromosome sizes varied from 3.0 to 4.4 µm (Figure 6b). Clusters of 45S rDNA were localized in the short arms of six SAT chromosome pairs 3, 4, and 7 of both genomes. Bright signals of 5S rDNA were observed in the pericentromeric regions of the short arms of chromosome pair 4 and also in the interstitial regions of the long arms of chromosome pair 8 of both genomes (Figure 6b).

Based on the chromosome morphology and distribution patterns of the revealed molecular cytogenetic markers, all chromosome pairs in karyotypes were identified, and chromosome karyograms and idiograms of *P. caeruleum* were constructed (Figure 6c–e).

### 2.4. Analysis of Leaf Epidermis

The analysis of leaves in *P. caeruleum* plants showed that epidermal cells had winding walls on both sides of the leaf, especially on the lower side (Figure 7a,b,d,e).

Above the veins, epidermal cells were elongated and had a rectangular spindle-shaped form. Glandular hairs were detected along the leaf veins and outgrowths of the epidermis along the edge of the leaves. On the lower side of the leaf epidermis, anomocytic stomata were presented, which were accompanied by three to six (rarely seven) epidermal cells (Figure 7b,e). 

Along the margin of both sides of the leaf, the epidermal cells were thick-walled and extended into papillae. In leaves of the colchicine-induced tetraploid plants, the predominance of large stomata (Figure 7e) was revealed unlike the control plants (Figure 7b).

Diploid plants had stomata with a smaller diameter as well as a lower frequency of stomata per unit leaf area compared to tetraploids (Table 2, Figure 7b,e). These results showed that stomata can be important indicators for the determination of ploidy level in *P. caeruleum*.

Glandular hairs are considered to be the diagnostic elements of leaves, and their form can be very different. In this study, glandular hairs were found along the leaf veins (Figure 7c,f) with a multicellular (two to four cells) stalk and a round or oval head. The hairs were thin-walled and sometimes had collapsed cells. 

## 3. Discussion

The successful induction of tetraploid plants usually depends on the effectiveness and toxicity of the applied chemical agents [35,36]. According to previous reports, the time of the colchicine exposure is positively correlated with seedling mortality in many plant species [27,37]. The increased lethality of the obtained plants after treatment with high concentrations of colchicine can be explained by the highly toxic effect of colchicine on the mitotic spindle (blocking spindle microtubules production) [38]. Therefore, the optimal concentration of the colchicine solution, as well as the duration of the treatment, is very important for the successful induction of polyploids. The results of our study showed that colchicine can be successfully used to induce polyploidy in the *P. caeruleum* cv. ‘Lazur’, and the largest number of autotetraploid plants (15%) was obtained after the treatment of seedlings with the 0.2% colchicine solution for 24 h.

Tetraploid plants of *P. caeruleum* were reported to have reduced fertility of pollen grains, which resulted in a smaller number of seeds [23]. In this study, tetraploid seeds were characterized by lower sowing qualities, such as germination energy (47%) and seed germination (55%), if compared with the control plants (78% and 96%, respectively). Despite tetraploid plants were shown to have relatively large seeds, anthers without pollen grains as well as sterile pollen grains were detected in such plants, which could lead to some decrease in the sowing qualities of their seeds [23,33]. In the subsequent generations of such plants; however, the differences in seed productivity between tetraploids and diploids could gradually level out [39].

Moreover, in colchicine-induced polyploids, various meiotic abnormalities can occur with the formation of uni-, tri- and tetravalents and, accordingly, with unequal chromosome segregation [27,32,33]. These disorders negatively affect the viability of pollen grains, and also the further process of double fertilization and formation of full-fledged seeds [40]. In the tetraploid plants studied in the present study, various chromosome configurations (bivalents, univalents, trivalents, and quadrivalents) were observed during the M-I stage, demonstrating the complexity of the synaptic configurations occurring during meiosis in these plants [41]. This can be attributed to the colchicine-induced chromosome doubling and formation of the autotetraploid with four sets of each chromosome in the genome. The problems in chromosome synapsis between the homologous chromosomes in autotetraploids might cause unbalanced chromosome segregation and result in high genetic instability in the pollen grains [41,42]. At the same time, the quadrivalent conjugations revealed in our study indicate the formation of balanced diploid meiotic products [43], therefore demonstrating the genome balance in the resulting tetraploids.

At the A-I stage of meiosis, a small number of cells with various abnormalities, including lagging chromosomes, bridges, and fragments, were revealed. Such anomalies indicate the incorrect orientation of the centromeres at the previous stages of meiosis and demonstrate the "mistakes" of different meiotic processes in plants [43]. These disorders can lead to the formation of gametes with the unbalanced number of chromosomes. The participation of such gametes in the fertilization process can result in the formation of aneuploid organisms, which in turn can decrease the practical value of tetraploids [33,44,45]. It has previously been reported that multivalents are disadvantageous due to their negative impact on fertility and karyotype stability [43], and this was confirmed by further selection for fertility results in fewer multivalents [33,43]. Nevertheless, high fertility has been reported to be correlated with increased quadrivalents at MI, suggesting that univalents and trivalents are the main cause of reduced fertility [46]. Therefore, the process of diploidization during meiosis could be a common strategy in the evolution of autopolyploid species [47]. This probably explains the abovementioned lower sowing qualities of seeds in tetraploids compared to the control diploids. In the following generations of such plants, however, mechanisms of meiosis optimization, and a gradual restoration of fertility can also occur [27,42]. In the present study, an increase in the percentage of meiosis disturbances was observed in colchicine-induced tetraploid plants when compared with the control diploids. Our findings are generally consistent with previously reported data on chromosome behavior during meiosis observed in other colchicine-induced tetraploid plants [27,42,44,48].

Our results on the karyotype characterization of diploid (2n = 2x = 18) and tetraploid (2n = 4x = 36) *P. caeruleum* are largely consistent with previously reported cytogenetic data (number of chromosomes and their sizes) obtained using monochrome chromosome staining [23]. At the same time, simple monochrome staining does not always allow clear identification of chromosomes in karyotypes of polyploids. Cytogenetic markers might provide valuable information about genome organization, characterize chromosome composition, and allow the identification of homologous pairs of chromosomes [49]. Eukaryotic 45S and 5S rDNA clusters are easily mapped by FISH on chromosomes in different species and widely used as chromosomal markers in plant karyological studies [50]. In the present study, for the first time, the karyotype of *P. caeruleum* was explored with the use of FISH with 45S and 5S rDNA probes, which made it possible to confirm the autotetraploid status of the colchicine-treated plants. FISH procedure revealed 45S rDNA clusters on three chromosome pairs (3, 4, and 7) in the control diploid plants and in six pairs (3, 4, and 7 in both genomes) in the obtained autotetraploids. In some karyotypes, small variations in the size of the 45S rDNA and 5S rDNA clusters were observed. It has previously been reported that the copy number of rRNA genes could slightly vary in different plant populations, which could be related to intraspecies diversity [51,52]. At the same time, considerable chromosome abnormalities such as chromosome rearrangements or aneuploidy were not revealed in the studied plants, indicating the relative stability of their karyotypes. The constructed chromosome karyograms of *P. caeruleum* can be useful for comparative karyotype analyses in different populations of this species as well as for further comparative karyological studies within the genus Polemonium useful for clarification of their phylogenetic relationships. However, more cytogenetic markers should be developed for this valuable species for further comprehensive studies of the blue cyanosis genome and also investigation of chromosomal and genetic diversity.

The polyploidy process can cause variations in plant growth and morphological traits including plant height, number of leaves, and number of side branches, as well as in the color and size of inflorescences [14,53]. Additionally, a correlation between the level of ploidy and physiological, morphological, and anatomical characteristics, such as plant height, color and shape of leaves and flowers, diameter of pollen grains, number of chloroplasts, size and density of stomata, has previously been described [54]. 

In the present study, for the first time, a microscopic analysis of the leaf epidermis was carried out in colchicine-induced tetraploid plants of *P. caeruleum*. The analysis of the leaf epidermis is considered to be an important technique for the standardization and quality control of medicinal plant materials [55,56]. Moreover, the structural characteristics of the leaf epidermis are used can be diagnostic features to confirm the authenticity of the aerial part of the plant and also for the rapid screening of tetraploid plants [57,58]. Some features of the leaf epidermis, such as the epidermal cell shape, stomata type, the presence or absence of pubescence, and cell wall thickness are useful tools for the identification of taxa and also their phylogenetic relationship with other taxa [57]. In this study, for the first time, common microdiagnostic features of *P. caeruleum* plants are revealed, which could be useful for the identification of its raw materials. These features included the presence of strongly sinuous cells in the leaf epidermis, anomocytic stomata localized on the lower epidermis of leaves, glandular hairs along the veins, and outgrowths of the epidermis along the edge of the leaves. However, in the colchicine-induced tetraploid plants, the stomata located on the lower leaf epidermis were larger if compared with the control plants. The larger stomata in the tetraploid plants compared to the diploids have previously been described in other plant species, including *Echinacea purpurea* [15], *Chamomilla recutita* [17], and the polyploids *Salvia* [59], *Tanacetun parthenium* [60] and *Thymus persicus* [61]. At the same time, several diploid plants with large stomata were also revealed, indicating the presence of natural variations in stomatal size in *P. caeruleum*.

Changes in the size and shape of various vegetative and reproductive parts were also reported for colchicine-induced tetraploid plants [62,63]. In particular, colchicine treatment caused a decrease in plant height [15,64,65]. In the present study, morphological differences between the control diploid and colchicine-induced tetraploid plants were easily identified visually, and these characteristics may be useful for preliminary identification of putative polyploids in *P. caeruleum* populations. In particular, in the first month after the treatment, growth retardation of the treated plants was observed, which could be related to the presence of residual colchicine having a damaging effect on the young tender seedlings. An initial decrease in growth rate has previously been described for colchicine-induced polyploid plants [15,41,60]. Finally, in the studied tetraploids, plant height was reduced (by 35%) compared to the control. At the same time, colchicine treatment led to an increase in the number of generative shoots, as well as leaf length and width, in the studied tetraploid plants (by 47%, 18%, 23% and 22%, respectively), which is important for obtaining greater amounts of raw materials, and, accordingly, triterpenoid saponin content compared to diploids. As reported previously, the biomass productivity depends on the organ structure, and also it is under the control of genome size [66]. Leaves in the tetraploid plants were dark green and also thicker than in control diploids of the same age. This is probably due to an increase in the number of chloroplasts, which was most likely accompanied by an increase in total chlorophyll content [35,61,67]. Additionally, flowering in the studied tetraploids occurred a week later compared to the control diploids, which could be related to the previously reported tendency of colchicine to turn on the gene(s) responsible for flowering induction by making the plant to respond to environmental signals such as photoperiodism [68].

In the obtained tetraploid plants, significant differences (*p* < 0.05) in terms of productivity, both raw materials and seeds, were revealed. In particular, the productivity of air-dried raw materials and the seed yield of tetraploids significantly exceeded that of the control (by 32% and 23%, respectively). Our results are generally consistent with previously reported data on induced tetraploid plants [14,67,69].

It has previously been shown that artificial polyploidy can increase the concentration of secondary metabolites in plants [15,70]. Moreover, a positive correlation has been reported between the total amount of genomic DNA and the content of secondary metabolites in medicinal raw materials [15,60,71]. In the studied tetraploid plants of *P. caeruleum*, increased content of biologically active compounds (triterpenoid saponins) was also revealed. Due to the duplication of the main genome in autopolyploid plants, such differences in the metabolic profile can result from changes in the mechanism that regulates the biosynthesis of individual compounds [28]. An elevated content of biologically active substances was revealed in the raw materials of some medicinal tetraploid plants, including *Echinacea purpurea* [15], *Artemisia annua* [18], *Hyoscyamus muticus* [20], and *Dracocephalum kotschyi* [53].

Thus, our findings demonstrated that tetraploid plants of blue cyanosis could be very useful in the cultivation of *P. caeruleum* for obtaining valuable high-quality raw materials of this medicinal plant, which is important for the pharmaceutical industry.

## 4. Materials and Methods

### 4.1. Plant Material

The seeds of *P. caeruleum* cv. ‘Lazur’ (K-26-3287, Russia) were obtained from the seed collection of the All-Russian Scientific Research Institute of Medicinal and Aromatic Plants (Moscow, Russia). The plants of *P. caeruleum*, grown from these seeds, were cultivated in the Botanic Garden of the All-Russian Institute of Medicinal and Aromatic Plants during three successive experimental seasons (summer 2019–summer 2021).

In late April 2019, seeds of *P. caeruleum* were germinated in Petri dishes on moist filter paper at room temperature (RT). Apical meristem of shoots of two- and four-leaf plants was treated with 0.2% aqueous colchicine (Fluka, Eindhoven, the Netherlands) for 24 h, and then the plantlets were washed with distilled water according to the previously described method [72]. Additionally, several concentrations of the aqueous solution of colchicine (0.05%, 0.1%, and 0.5%) were examined. Thirty days after the treatment, the effect of various concentrations of colchicine on plant survival was evaluated. The control plants were treated with water. In each variant of the experiment, 500 seeds were used. Plant seedlings treated with colchicine (C_0_ generation) and also the control seedlings were planted in a greenhouse. 

The control and colchicine-treated plantlets (at the stage of 6–8 leaves) were visually analyzed to identify distinctive morphological features, and they were also tested for the number of chromosomes in karyotypes. Then, they were planted in the field in a crop geometry of 60 cm × 30 cm according to the previously described approach [73]. The revealed tetraploid plants were isolated with parchment insulators to avoid cross pollination influence.

The soil cover of the trial plots was represented by sod-podzolic (moderately podsolized) silt loams (80–100 cm) underlain by moraine deposits. The plough layer (22–23 cm) was brownish-grey and had a crumbly structure and a sandy loam soil composition. The content of healthy water-stable aggregates (>0.5 mm) was 40–50%.

The soils of the trial plots had the following nutritional characteristics: humus content 2.1%, pH 5.5, the content of mobile phosphorus (P_2_O_5_)-52 mg/kg and exchangeable potassium (K_2_O)-87 mg/kg. In primary cultivation, fertilizer treatment with N_30_P_30_K_30_ was performed according to the standard techniques described previously [74].

### 4.2. Agro-Morphological Characterization

In the first vegetation year (2019), the colchicine-treated plants went into the winter in the rosette stage. In the second (2020) and third (2021) seasons, the colchicine-treated plants went through a full cycle from the growth of a secondary rosette of leaves to seed (C_1_ and C_2_ generations, respectively) maturity. Plant vegetative parameters, including plant height, number of generative shoots, stem thickness, leaf length and width, were measured in ten randomly selected plants from each trial plot in a mass flowering period (2020–2021) according to the standard techniques [75]. At least 50 plants were analyzed from each population. To evaluate the productivity of raw materials, the roots of plants were dug up at the end of the growing season (late September 2021), washed with running water, and dried at 40 °C for 72 h. The yield of seeds was harvest at the maturity stage (in September 2020 and 2021) [73]. Germination energy and seed germination were determined for the C_1_ seeds and several of the normally germinated seeds (expressed as a percentage of the total number of seeds taken for germination) on the seventh day and tenth or eleventh days of germination, respectively. Statistical data analysis was performed using standard functions of Microsoft Excel 2013. 

### 4.3. Determination of the Content of Saponin in Plant Raw Materials

In September 2021, plant rhizomes with adventitious roots as well as the rosettes of leaves were collected, air-dried and ground into a powder. The content of triterpene saponins in terms of β-escin was determined in these raw materials according to the previously described method with minor modifications [76]. Preparation of solution “A”: 0.5 g of the crushed raw materials were placed into a cartridge and purified with chloroform in a Soxhlet extraction apparatus (Bionics Scientific Technologies, Delhi, India) for 2 h. The cartridge with the raw materials was transferred to the conical flask. Then, 50 mL of 70% ethanol was added, and extraction was performed using a water bath for 1.5 h. Then, 5 mL of the filtrate was distilled to dryness using a rotary evaporator. The dry residue was dissolved in glacial acetic acid and transferred into a 25 mL volumetric flask. Preparation of solution “B”: 1.0 mL of solution “A” was placed in a volumetric flask; the total volume was adjusted to 25 mL with glacial acetic acid and stirred. Then, 2.0 mL of solution "B" was placed in a flask, then 2 mL of glacial acetic acid and 2 mL of concentrated sulfuric acid were added and boiled with the reverse cooler for 1 h. After that, the solution was cooled, and the optical density was determined using a scanning spectrophotometer SF-104 (Akvilon, Moscow, Russia) at a wavelength of 282 nm. As a reference solution, the solution containing of 4 mL of glacial acetic acid and 2 mL of concentrated sulfuric acid (kept under the same conditions) was used. As a control, the optical density of the standard sample of β-escin (DeltaOrigin, Moscow, Russia) prepared under the same conditions was used.

### 4.4. Analysis of Leaf Epidermis

Epidermal peel preparations were made from the rosettes of leaves according to the standard technique [77]. Dry leaves were boiled in 3% NaOH solution for 3 min. The leaves were washed with distilled water for 5 min. After that, the leaf epidermal tissue was transferred onto the slide, added a drop of a solution containing ethanol:glycerol:chloralhydrate (1:1:1) and covered with a cover glass. The stomata characteristics were estimated in randomly selected 10 plants of diploid and tetraploid *P. caeruleum*. The sample preparations were analyzed using the Altami BIO 2 LED biological microscope, which was equipped with the 3.1 Mp digital eyepiece USB camera (Altami, St. Petersburg, FL, USA). The images were processed with Adobe Photoshop CS3 (Adobe, Birmingham, AL, USA) software. At least 15 images were examined for each sample. Statistical data analysis was performed using standard functions of Microsoft Excel 2013.

### 4.5. Chromosome Spread Preparation

For chromosome counting, in the rosette stage of the plants, tips of actively growing roots (0.5–1 cm) were placed in 1% acetocarmine solution in 45% acetic acid for 15–20 min at room temperature (RT). Then, the roots were transferred to the slide, and the root meristem was cut with a preparation needle and placed in a drop of 1% acetocarmine solution and covered with a coverslip. After that, chromosome spreads were prepared and chromosome numbers in the cells were immediately analyzed.

For fluorescent in situ hybridization (FISH) assays, the chromosome spread preparations were made as described previously with minor modifications [52]. Seeds were germinated in Petri dishes on the moist filter paper at RT. Root tips (0.5–1 cm) were excised and put into ice-cold water for 16–20 h for accumulation of mitotic divisions. Then, the root tips were fixed in ethanol: acetic acid (3:1) fixative for 48 h (RT). The roots were stored in 1% acetocarmine solution in 45% acetic acid for 15 min. One root was placed on the slide, the root meristem was cut from the root cap, and a squashed preparation was made with the use of a cover slip. After freezing in liquid nitrogen, the cover slip was removed, and the slide was dehydrated in 96% ethanol and air dried.

Meiotic chromosome preparations were made according to the previously described technique with minor modifications [52]. Young floral buds (prefoliation) were put into ethanol:acetic acid (3:1) fixative for 30 min at 4 °C. The fixed buds were placed into the 1% acetocarmine solution in 45% acetic acid for 15–20 min at RT and then transferred on the slide. The anthers were taken out of the flower buds, cut with a preparation needle in a drop of 45% acetic acid, covered with a cover glass, and chromosome spreads were prepared. After freezing in liquid nitrogen, the cover glasses were removed; the slides were dehydrated in 96% ethanol for 3 min and air dried for 15 min.

### 4.6. FISH Procedure 

For the FISH procedure, two wheat DNA probes pTa71 containing 18S-5.8S-26S (45S) ribosomal DNA (rDNA) [78] and pTa794 containing 5S rDNA sequence [79] were used. These DNA probes were labeled directly with fluorochromes Aqua 431 dUTP and Red 580 dUTP (ENZO Life Sciences, Farmingdale, NY, USA) by nick translation according to the manufacturer’s protocols. FISH procedure was performed as described previously [80], with some modifications. Before FISH, chromosome slides were pretreated with RNase A (Roche Diagnostics, Mannheim, Germany) dissolved in 2 × SSC (1 mg/mL) at 37 °C for 1 h. Then, the slides were washed three times for 8 min in 2 × SSC, dehydrated through an ethanol series (70%, 85%, and 96%) for 2 min each and air dried for 20 min. An amount of 15 μL of hybridization mixture containing 40 ng of each labeled probe was added to each slide. Then, the slides were covered with coverslips, sealed with rubber cement, denatured at 74 °C for 5 min, chilled on ice and placed in a moisture chamber at 37 °C for 16 h. Then, the slides were washed with 2 × SSC (three times for 10 min each at room temperature), dehydrated through a graded ethanol series and air dried. Then, 15 µL of hybridization mixture containing 40 ng of each labeled probe was added to each slide. The slides with DNA probes were covered with coverslips, sealed with rubber cement, denatured at 74 °C for 5 min, and placed in a moisture chamber at 37 °C. After overnight hybridization, the slides were washed with 0.1 × SSC (10 min, 44 °C), twice with 2 × SSC (10 min at 44 °C), followed by a 5 min wash in 2 × SSC and two 3 min washes in PBS at room temperature. Then, the slides were dehydrated, air dried and stained with DAPI (4′,6-diamidino-2-phenylindole) dissolved (0.1 μg/mL) in Vectashield mounting medium (Vector Laboratories, Burlingame, CA, USA).

### 4.7. Analysis of Chromosomes

Chromosomal slides were analyzed using the Olympus BX-61 epifluorescence microscope (Olympus, Tokyo, Japan). Chromosome images were captured with monochrome charge-coupled device camera (Cool Snap, Roper Scientific, Inc., Tucson, AZ, USA). Then, they were pseudo-colored and processed using Adobe Photoshop 10.0 (Adobe, Birmingham, CA, USA) software. At least 15 diploid and 15 tetraploid plants (15 metaphase plates for each plant) were analyzed. For analysis of meiosis, 10 plants (50 meiotic cells for each plant) were examined. Chromosome pairs in karyotypes were identified according to chromosome size and morphology, as well as the localization of the chromosome markers. In the karyograms, chromosome pairs were set in the decreasing order of size.

## Figures and Tables

**Figure 1 plants-11-02585-f001:**
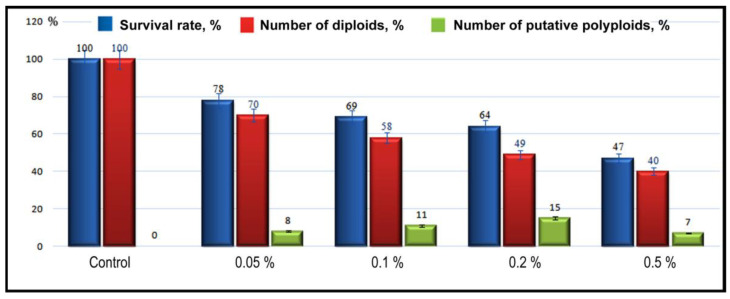
Influence of different concentrations of colchicine on the survival rate and polyploidy induction in *Polemonium caeruleum*.

**Figure 2 plants-11-02585-f002:**
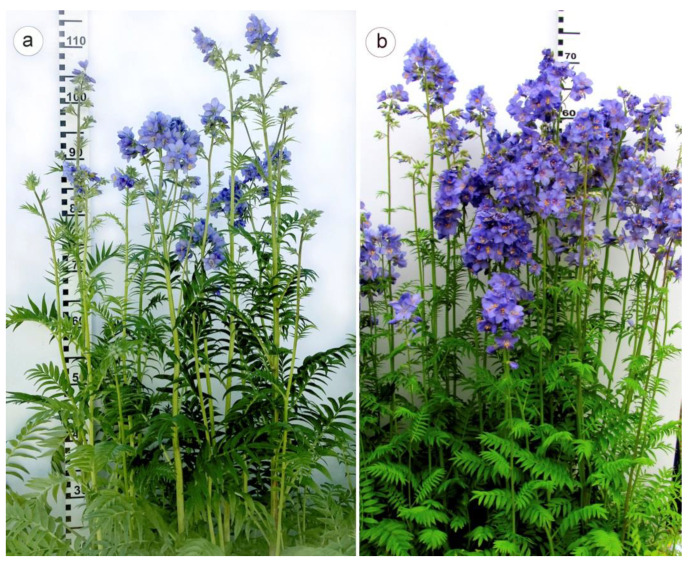
Flowering control diploid (**a**) and colchicine-induced tetraploid (**b**) plants of *Polemonium caeruleum*.

**Figure 3 plants-11-02585-f003:**
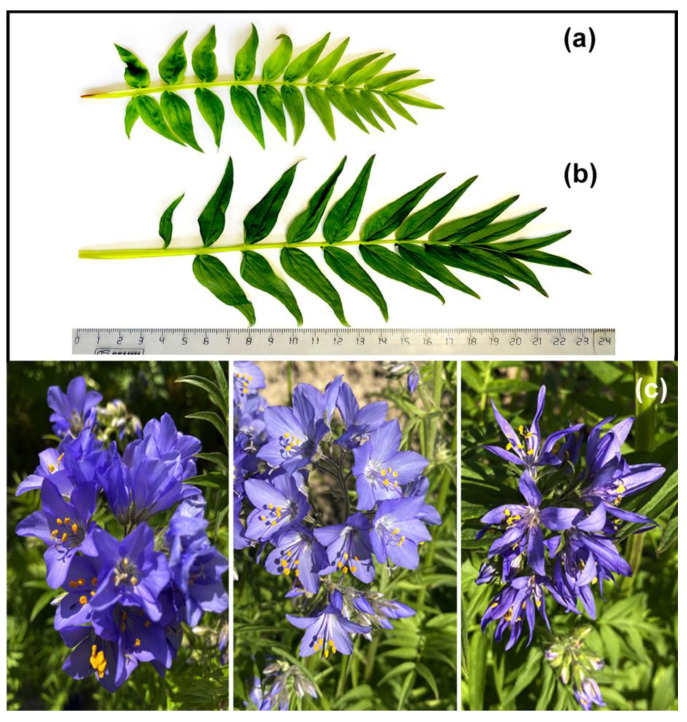
Leaf blades of the diploid (**a**) and tetraploid (**b**) plants of *Polemonium caeruleum;* the diversity in the color and shape of inflorescences observed in the tetraploids (**c**).

**Figure 4 plants-11-02585-f004:**
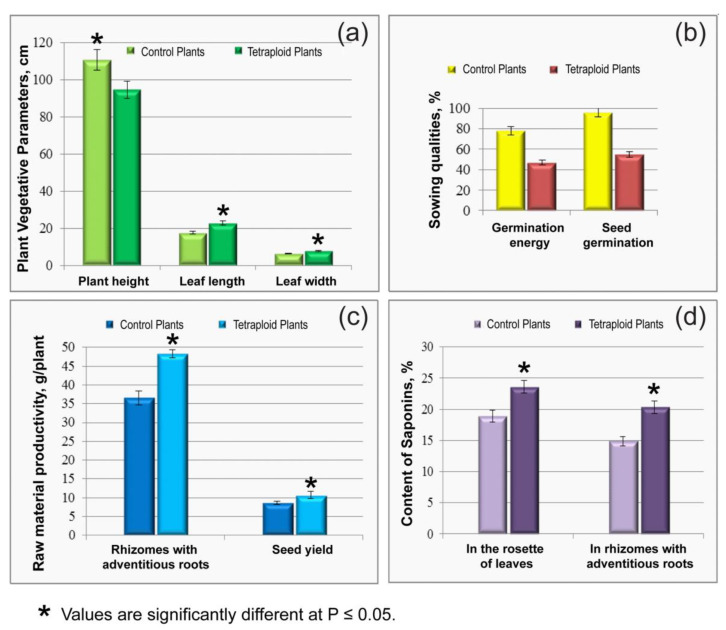
Vegetative parameters: Plant height, Leaf length, Leaf width (**a**); Sowing qualities (**b**); Raw material productivity (**c**) and also Content of saponins (**d**) in the control and colchicine-induced tetraploid plants of *Polemonium caeruleum.* * Values are significantly different at *p* ≤ 0.05.

**Figure 5 plants-11-02585-f005:**
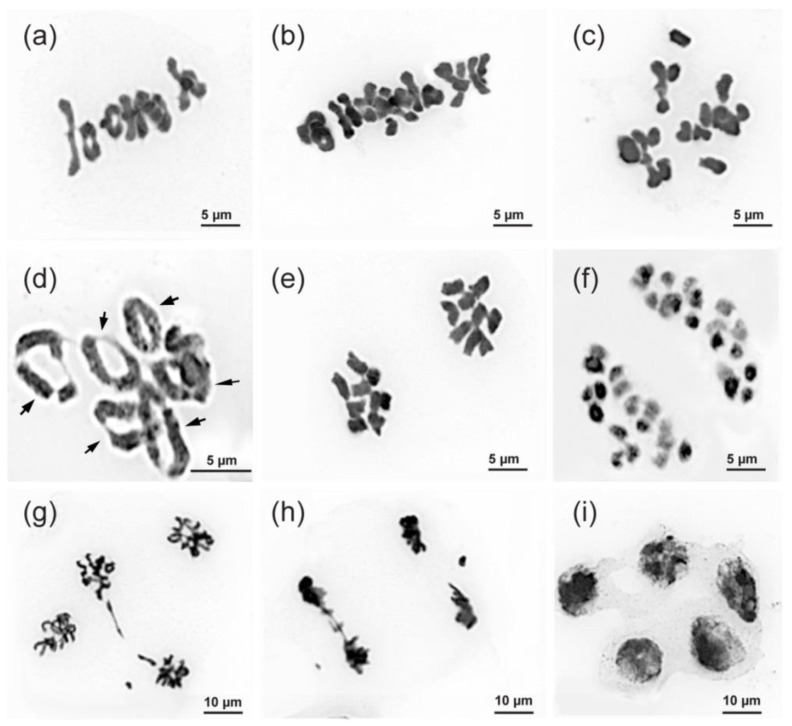
Chromosome behavior during meiosis in maternal pollen cells of *Polemonium caeruleum*. (**a**) M-I, a diploid cell (9^II^); (**b**) M-I, a tetraploid cell (18^II^); (**c**) M-I, chromosome lagging in a diploid cell; (**d**) M-I, quadrivalents are indicated with arrows; (**e**) A-I, (9:9); (**f**) A-I, (18:18); (**g**) A-II, perpendicular spindle orientation in a tetraploid cell; (**h**) A-II, chromosome elimination in a tetraploid cell; (**i**) pentads. Scale bars are specified under the images.

**Figure 6 plants-11-02585-f006:**
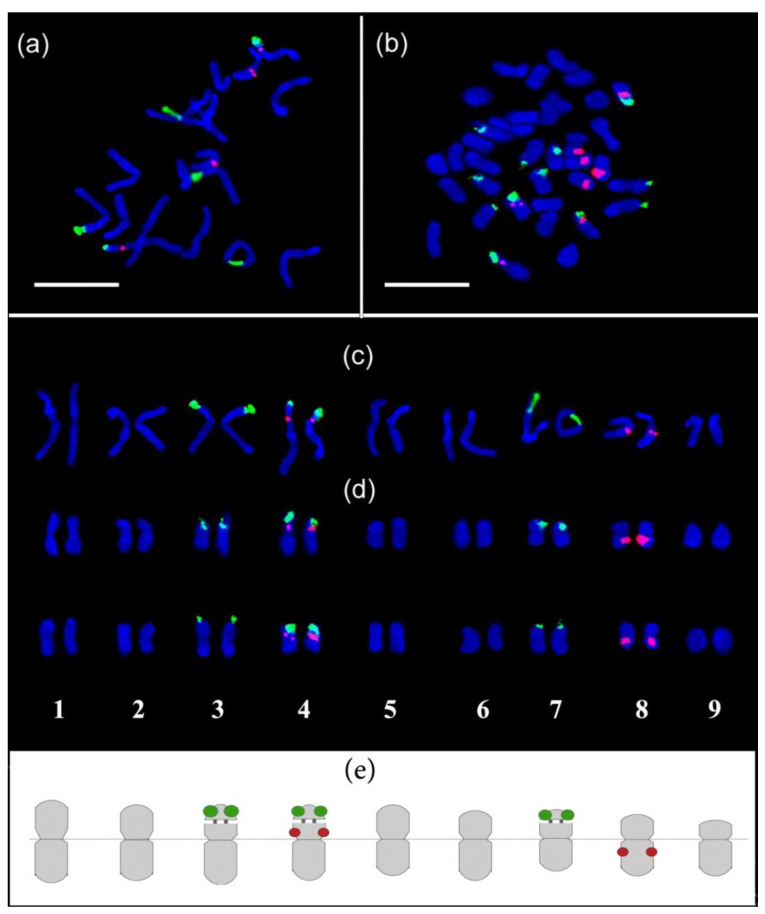
Chromosome spreads and karyograms of the studied diploid (**a**,**c**) and tetraploid (**b**,**d**) plants of *Polemonium caeruleum* after FISH with 45S rDNA (green) and 5S rDNA (red). Idiograms of diploid *P. caeruleum* chromosomes showing 45S (green) and 5S (red) rDNA sites (**e**). Scale bar—5 µm.

**Figure 7 plants-11-02585-f007:**
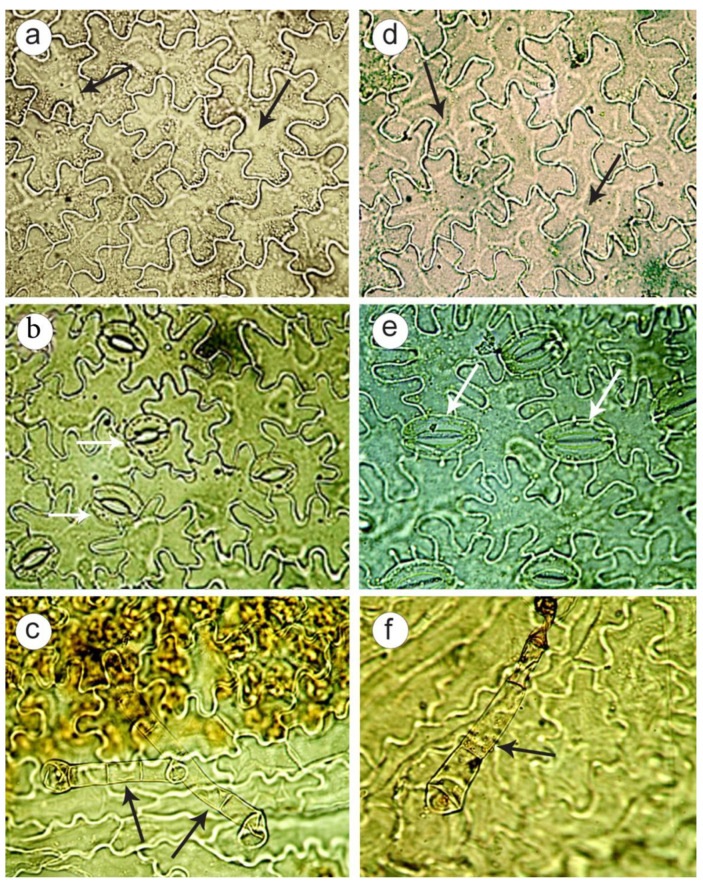
Comparative characterization of leaf epidermis in the diploid (**a**–**c**) and tetraploid (**d**–**f**) plants of *Polemonium caeruleum* L. (**a**,**d**) Cells with winding walls on the upper epidermis of a leaf (×400; black arrows); (**b**,**e**) cells with winding walls and stomata on the lower epidermis of the leaf (×400); white arrows; (**c**,**f**) glandular hairs along the leaf vein (×400; black arrows).

**Table 1 plants-11-02585-t001:** Effects of colchicine treatment on morphometric parameters and productivity of plants in *Polemonium caeruleum*.

Plant Vegetative Parameters	Control Plants	Tetraploid Plants
Plant height, cm	110.8 ± 3.31 *	73.2 ± 2.31
Numbers of generative roots	6.1 ± 0.50	7.2 ± 0.88 *
Stem thickness, cm	0.68 ± 0.064	1.00 ± 0.094 *
Leaf length, cm	17.7 ± 1.11	22.8 ± 0.44 *
Raw material productivity (rhizomes with adventitious roots), g/plant	36.5 ± 3.33	48.3 ± 4.71 *
Seed yield, g/plant	8.7 ± 0.56	10.7 ± 1.28 *
Weight of 1000 seeds, g	1.46 ± 0.116	1.54 ± 0.125
Germination energy, %	78	47
Seed germination, %	96	55
**Content of Saponins, %**		
In the rosette of leaves	18.89 ± 1.37	23.58 ± 2.05 *
In rhizomes with adventitious roots	14.92 ± 1.24	20.36 ± 1.23 *

* *Values are significantly different at p* ≤ 0.05.

**Table 2 plants-11-02585-t002:** Stomata characteristics in diploid and tetraploid Polemonium caeruleum.

Characteristic	Diploid (Mean)	Tetraploid (Mean)
Stomata length (µm)	11.70 ± 1.12	15.30 ± 1.25 *
Stomata width (µm)	6.83 ± 0.24	7.10 ± 0.17 *
Stomata frequency/(mm)^2^	83.7	85.2

* *Values are significantly different at p* ≤ 0.05.

## Data Availability

All data generated or analyzed during this study are contained within the article.

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
