# Peer review of "Agro-Morphological and Cytogenetic Characterization of Colchicine-Induced Tetraploid Plants of Polemonium caeruleum L. (Polemoniaceae)"

_plants, 2022, doi:10.3390/plants11192585_

Round 1
Reviewer 1 Report
Comments
on the manuscript entitled” Agro-Morphological and Cytogenetic Characterization of Col- chicine-Induced Tetraploid Plants of Polemonium caeruleum L. (Polemoniaceae) by Tatiana E. Samatadze et al.
The above manuscript provides convincing data set about the morphological, the cytological features, furthermore the raw material productivity and the content of triterpene saponins in the tetraploid Polemonium caeruleum L.plants with (2n = 4Ñ… = 36) plants in comparison to the diploid variants. The later parameters were significantly higher than in the control diploids. In tetraploid plants, the raw material productivity and seed yield exceeded (by 32% and 23%, respectively) those parameters in the control plants. The authors provide chromosome karyograms and idiograms and changes in characteristics of the epidermal cells. Considering the presented data and the quality of this work I recommend the publication.
Some suggestions:
1. The authors can discuss the increased root size as it was shown earlier in tetraploid willows ( Dudits et. al 2016 Plant Physiology 170, pp. 1504–1523).
2. Furthermore, more attention could be devoted to the increased triterpenoid saponin content.
In summary I recommend the publication.
Author Response
Dear Editorial Board Team,
Thank you for your email dated 26 September 2022 enclosing the decision and the reviewers’ comments on Manuscript plants-1945432
We are very grateful to the reviewers’ comments and thoughtful suggestions which are valuable in improving the quality of our manuscript. We have revised the manuscript accordingly, and our responses are given below. Besides, the manuscript with tracked changes has been applied. Changes to the manuscript are shown in yellow and green.
Sincerely yours,
Alexandra V. Amosova
Corresponding author
Response to the Reviewer's Comments and Suggestions
Reviewer 1: on the manuscript entitled” Agro-Morphological and Cytogenetic Characterization of Col- chicine-Induced Tetraploid Plants of Polemonium caeruleum L. (Polemoniaceae) by Tatiana E. Samatadze et al.
The above manuscript provides convincing data set about the morphological, the cytological features, furthermore the raw material productivity and the content of triterpene saponins in the tetraploid Polemonium caeruleum L.plants with (2n = 4Ñ… = 36) plants in comparison to the diploid variants. The later parameters were significantly higher than in the control diploids. In tetraploid plants, the raw material productivity and seed yield exceeded (by 32% and 23%, respectively) those parameters in the control plants. The authors provide chromosome karyograms and idiograms and changes in characteristics of the epidermal cells. Considering the presented data and the quality of this work I recommend the publication.
Answer: We thank the reviewer for valuable and constructive comments and suggestions very much.
Some suggestions:
- The authors can discuss the increased root size as it was shown earlier in tetraploid willows ( Dudits et. al 2016 Plant Physiology 170, pp. 1504–1523).
Answer: The increased size of roots and leaves has been discussed. Lines 338-341
- Furthermore, more attention could be devoted to the increased triterpenoid saponin content.
Answer: The increased triterpenoid saponin content has been discussed. Lines 338-339
Reviewer #2:
Comments and Suggestions for Authors
In this manuscript, tetraploids of Polemonium caeruleum were induced with colchicine, and agro-morphological and cytogenetic characterization were compared with diploid plants, which demonstrated that polyploid breeding is useful for improvement of Polemonium caeruleum. However, several details need to be revised in this manuscript.
Answer: We are very grateful to the reviewer’s comments and thoughtful suggestions.
- Because the Results part is the second part, before of the M&M part, the authors need give a short description on the methods of polyploid induction in the 2.1 part.
Answer: The short description on the methods of polyploid induction has been discussed in the Result section. Lines 90-92
- In the 2.1 part, for the putative plants, how many tetraploids were detected by chromosome counting? In general, the treatment of apical meristem of shoots with colchicine is possible to produce mixoploids. Was flow cytometry used to analyze the ploidy level of treated plants? If possible, I suggest that the authors can provide a flow cytometic figure of putative plants.
Answer: The number of the detected tetraploids was specified. The revealed mixoploid root tissues were discarded from the experiment as they had unstable polyploid nature. The ploidy level was determined cytogenetically, and therefore, the flow cytometry method was not used in the experiments. Lines 106-112
- Figure 1, what do the red and green bars represent? Percentage or number?
Answer: In Figure 1, the red and green bars represent the number of diploids (%) and number of putative polyploids (%), respectively, that shown above the histograms.
- In Figure 4, what is the unit of y axis in drawing a? And the drawing b should be same with others in size.
Answer: It is misprint (cm). Corrected.
- Are there any significant difference on germination engerge and seed germination rate between control and tetraploid plants? What method was used to analyze the difference?
Answer: Germination energy and seed germination were determined for C1 seeds as a number of the normally germinated seeds (expressed as a percentage of the total number of seeds taken for germination) on the seventh day and tenth-eleventh days of germination, respectively (specified in MM section, lines 410-413). In this study, we revealed that germination energy and seed germination rate were lower in tetraploids compared to the control diploids. The possible significance of this difference will be determined in our further studies on the study of next generation development of tetraploid P. caeruleum.
- Where is table 2?
Answer: It is a misprint. Corrected.
- For a research article, excessive references were cited.
Answer: Some references were deleted.

Reviewer 2 Report
In this manuscript, tetraploids of Polemonium caeruleum were induced with colchicine, and agro-morphological and cytogenetic characterization were compared with diploid plants, which demonstrated that polyploid breeding is useful for improvement of Polemonium caeruleum. However, several details need to be revised in this manuscript.
1. Because the Results part is the second part, before of the M&M part, the authors need give a short description on the methods of polyploid induction in the 2.1 part.
2. In the 2.1 part, for the putative plants, how many tetraploids were detected by chromosome counting? In general, the treatment of apical meristem of shoots with colchicine is possible to produce mixoploids. Was flow cytometry used to analyze the ploidy level of treated plants? If possible, I suggest that the authors can provide a flow cytometic figure of putative plants.
3. In Figure 1, what do the red and green bars represent? Percentage or number?
4. In Figure 4, what is the unit of y axis in drawing a? And the drawing b should be same with others in size.
5. Are there any significant difference on germination engerge and seed germination rate between control and tetraploid plants? What method was used to analyze the difference?
6. Where is table 2?
7. For a research article, excessive references were cited.
Author Response

(The authors gave the same response as above.)
